# Religions, Women and Discourse of Modernity in Colonial South India

**M. Christhu Doss**

School of Social Sciences, Christ University, Bengaluru 560029, India; christhu.doss@christuniversity.in

**Abstract:** Colonial education and missionary discourse of modernity intensified struggles for continuity and change among the followers of Hinduism and Christianity in nineteenth century India. While missionary modernity was characterised by an emphasis on sociocultural changes among the marginalized women through Christian norms of decency, orthodox Hindus used traditional cultural practices to confront missionary modernization endeavours. This article posits that the discourse of missionary modernity needs to be understood through the principles of Western secular modernity that impelled missionaries to employ decent clothing as a symbol of Christian femininity. It argues that missionary modernity not only emboldened the marginalized women to challenge their ascribed sociocultural standing but also solidified communitarian consciousness among the followers of Hinduism and Christianity substantially. Even though Travancore state defended the entrenched customary practices, including women's attire patterns, with all its potency through authoritative proclamations, it could not dissuade missionaries from converting the marginalized women to missionary modernity.

**Keywords:** colonialism; Christianity; gender; Hinduism; missionary modernity; Nadars; Nairs; tradition

## 1. Revisiting the Binary

The intricate interconnectedness between women missionaries and discourse of modernity in nineteenth-century India needs to be re-examined historically and critically in light of tradition-modernity binary (D. M. Menon 2002, pp. 1662–63). Influenced by the principles of secular modernity, missionaries identified themselves with and fostering the ideas of equality and respectability. In South India, Travancore in particular, the ideas of tradition and modernity were often perceived as polar contraries. Driven mainly by religious intents, missionary modernity struggled to transform the marginalized women through the Christian notion of decency. The defenders of tradition strived to retain some cultural practices of Hinduism in order to validate orthodox femininity in the nineteenth century (Gusfield 1967; Bendix 1967; Kothari 1968). When missionaries perceived the emerging Hindu cultural assertions as a serious impediment to their modernisation endeavours, they encouraged the marginalized women to confront traditional taboos including attire pattern (Kent 2004, pp. 54–77). The discourse of missionary modernity in Travancore was thus centred on the utilitarian aspects of colonial education, proselytization and public decency that emphasized superficial changes in the cultural life of the people, particularly with regard to clothing.

The missionary perception that Christians should never be 'filthy' or 'indecent' became an essential component of proselytization. Even though the Christian notion of decency was not made to be a part of the formal missionary pedagogy, it eventually became the unwritten syllabus of instruction in their educational institutions. Missionaries continued to believe that religious instruction and decency are closely linked to each other. They argued that it was Brahmanical Hinduism that divided people on caste lines and disseminated the ideas of purity and pollution. Hence, missionaries became critical towards the institution of caste itself that kept the marginalized women in a state of seclusion socially and culturally.[1]

Marginalized communities used education and conversion to Christianity as tools to challenge social and cultural hierarchy in the nineteenth century. As Uma Chakravarty argues, women were desperate in breaking their prison-houses largely through missionary education that provided the converts particularly from the lower castes with a means to access employment opportunities. This enhanced the possibilities of upward mobility though on a selective basis. Changing new identities impelled people from the lower castes challenge the symbolic world of status and privilege. For example, the struggle for attire decency in Travancore not only increased the numerical strength of Christian converts among the Nadars, but also propelled the women to fight for the same status, same modesty and same privilege to cover the breast in the same way as upper caste women did. Even though the very right to be 'modest' itself was generally attributed to caste status, Nadars went beyond missionary notions of decency in some senses (Chakravarti 2018).

The caste complexity in Travancore society not only enforced marginalized women to divulge their breasts but also exposed them to mental torment and psychological nuisance (Shaji 2017, p. 11). Missionaries were seemed to be radical than the colonial political administrators in challenging and changing believes, practices and customs of the Hindus (Beidelman 1982, pp. 2–16; Cox 1994). Covering female body appears to have equal hitches as uncovering of the body. Generally, the patriarchal society believed that the act of covering the female bodies was a form of dignity of female sexuality within the framework of family. This belief system provided men a kind of ownership right over women's bodies. Covering the body was also seen as an indication of the property right of the husband—the Nadar men (earlier Shanars) in this context. As Sonja Thomas argues, one of the most conspicuous features of the marginalized women in Travancore was the practice of bare-breastedness. Brahmin pundits in the nineteenth century had not only interpreted Hindu Smritis to justify the deep-rooted practice of bare-breastedness but also vindicated sexual ownership of the marginalized women. Consequently, the supposedly lower caste men and women were not permitted to cover their bodies in front of orthodox men and women (S. Thomas 2018, pp. 46–51).

The origin of the orthodox femininity discourse can be traced back to the ancient Indian lawgiver Manu who specified that a woman's father should protect her in childhood, her husband in her youth and her sons in her old age. For example, Indian feminist Pandita Ramabai argued that it was Manu who had created a cultural ecosystem where women were not only overlooked but also were seen as dependent creatures that can never be independent in thinking and action. This patriarchal perception continued during the early modern and colonial periods particularly, until the first half of the nineteenth century India where majority of the women hardly had access to education (Sarasvati 1888, pp. 50–60; Mondal 2017).

The discourse of missionary modernity was instrumental in making the nineteenth century India to be a battlefield for ideologies that addressed questions of identity, decency, femininity and respectability (Sen 2015). During this period, the question of how to represent the cultural past of a community was seen as a much-debated political, as well as cultural, issue. In Travancore, the controversy was characterised particularly by the discourse of tradition-modernity (Gottlob 2002, pp. 75–97; 2011, pp. 6–8). As Ranajit Guha argues, the British colonizers were keen on dislocating the people in India politically, ethnically, materially, culturally and religiously. The British officials and missionary agencies portrayed themselves as a 'superior' race, which followed higher ideals of Christianity. On the contrary, the colonized Indians were depicted as impoverished, 'inferior', 'superstitious' and 'barbarous' people (Guha 1997, pp. 1–3).

It is significant to note that postcolonial writings on social change scarcely used terms such as 'civilized' or 'barbarism' directly. Instead, these writings consciously employed usages such as modernity and tradition to give an impression of being unbiased when appraising the progress of individuals, communities and nations based on Western standards (Huntington 1968). However, the term modernity, as Dilip Menon contends, "comes to us masking both its origins within a distinct geographical space as well as an imagina-

tion almost entirely concerned with a description of change in the west." This suggests that modernity in the nineteenth century south India was neither a temporally nor a geographically grounded phenomenon. Even though there was an increasing suspicion towards the applicability of modernity as a natural historical process, colonial government and missionaries strived to make it appear as a natural historical process in South India (D. M. Menon 2002, pp. 1662–67).

If there was one particular narrative that dominated the entire colonial literature in nineteenth-century India, it was the discourse of change, commonly referred to as modernity. Partha Chatterjee argues that, in the nineteenth century, everything was changing, and nothing remained the same. The colonial government and missionaries were keen on altering the very foundations of Indian cultural ethos by creating a discourse of modern consciousness of the self (Chatterjee 1993, pp. 49–50 and 117–19). The missionary discourse of modernity led to a series of contestations and transfigurations in the spheres of education, religion, governance, body and public life. It often challenged the manner in which social affairs were conducted, though the modern way of doing things was not written on a clean slate (Kaviraj 2000). The impact of colonialism on Indian society, culture, mind and body was so deep and incomprehensible that it led to what Ashis Nandy called 'the psychological uprooting and cultural disruption' (Nandy 1983, pp. 51–52). The British colonists and missionaries endeavoured to break down the entire cultural framework of the Indian society through their civilizing mission (Marx 1853, p. 652; Srinivas 1962, pp. 2–18; Rudolph 1965, pp. 976–77). The Hindus were depicted as heathens, fornicators, idolators, adulterers, covetous, drunkards and extortioners.[2] Thus, missionary modernity was perceived as an ideological instrument for creating social and cultural changes (Shils 1961, pp. 3–9; Rudolph 1965, pp. 982–83).

The principles of Western secular modernity and modern institutional structures created various forms of transformation and cultural dislocation. The cultural dislodgment was conspicuous as there were perceptible vagaries in religious ideals, customs, attire and profession (Nandy 1983, p. 9). As a robust upholder of tradition, Travancore state played a crucial role in asserting its deep-rooted customs through proclamations that created upheavals among the marginalized sections. In fact, in some of his paintings, T. Murali, a twentieth century Kerala-based artist depicts the cultural complexities of Travancore state. In his painting on what he called the Shanar revolt that took place in the 1850s, he demonstrated how there were instances where the marginalized women with the help of missionaries asserted their rights to cover their upper body with clothes of their choice. His portrayal of Nangeli, an Ezhava woman who chopped off her breasts in protest against *mulakkaram* or breast tax imposed by the Travancore state, reveals how marginalized women experienced different forms of injustice that were directly linked to questions of identity, psychic and emotions (Nazeer 2015; Pan 2021). It also demonstrates how Travancore state was assertive in imposing dissimilar codes of conduct to be followed by women. It was women missionaries who were instrumental in challenging the ideology of the Travancore state and confronting the deep-rooted orthodox femininity. Even though the discourse of missionary modernity strived to transcend boundaries of religions, it was limited largely to those women who either converted to Christianity or to those women who expressed their willingness to be moulded by missionary ideals. Missionaries argued that the transition from orthodox femininity to Christian femininity is possible only through Christian norms of domesticity and decency. While divulging the upper part of female body was perceived to be a symbol of orthodox femininity, the very idea of covering the body was equated with the standards of Christian femininity.

The British Residents and missionaries in Travancore assiduously endeavoured to liberate the marginalized women from the orthodox femininity and other inherently overbearing cultural traditions of Travancore. They claimed that the atrocities perpetrated on women could be challenged only through an orderly, lawful, moral, civilizational and rational procedure of governance. As Partha Chatterjee argues, the practical implication of the

criticism of Hindu traditions and customs was essentially a colonial project of modernizing people in general and marginalized women in particular (Chatterjee 1997).

If the battles of Plassey and Buxar triggered the modernizing project of the British in the 1750s and 1760s, missionaries, liberals and the defenders of Western secular modernity through a superior racial and religious ideology further intensified the process missionary modernity. While modernization advocated a wider materialistic change at the institutional and personal level, Westernization believed in the idea that Western culture was the epitome of a good lifestyle that involves a drastic change in the behavioural aspects that are related to customs, including food and attire. In some instances, missionaries used both modernization and Westernization interchangeably to validate the claims of religious superiority in the nineteenth century. They used religion as an instrument of moral and material advancement in social structures, in minds, and in bodies of the colonized people. Missionaries believed that the material benefits of Western science, the advantages of economic progress, the cultural benefits of Western education and the moral benefits through Christian ethics would facilitate the Indian women to conform to Victorian norms of respectable dress, public decency, sexuality and family life (Narayan 1995). Thus, missionaries attempted to replace orthodox femininity with a Christian femininity.

As Ranajit Guha maintains, the colonial government and missionaries collectively sought to improve moral and material conditions of the people through social, cultural, educational, legal and ecclesiastical regulations (Guha 1997, pp. 33–34 and 80–81). As apologists of European civilization, a considerable number of European intellectuals argued that in an age of optimism British rule would be a necessary evil. They maintained that the colonizers were not a group of self-seeking, insatiable, ethnocentric vandals and self-chosen carriers of what Ashis Nandy says a cultural pathology, but ill-intentioned, defective historical instruments that instinctively worked for the upliftment of the neglected and deprived sections (Nandy 1983, pp. 33–35).

Missionaries provided greater security to those converted Christians, who had been disturbed by private persecution and legal spoliation. They were persuasive in their approach, which was applauded by some governor generals. For example, Governor General William Bentinck who had cordial relationship with missionaries remarked that the profession of missionaries was a holy one, which he felt was most gratifying. He specified that the sole objective of missionaries was religious conversion, whereas the fundamental principle of British rule in India was strict religious neutrality. Nevertheless, he ventured to give it as his firm option that in all the schools and colleges under the support of the government, the principle of religious neutrality cannot be too strongly enforced. He also advised the missionaries to expand the scope of education by all means as a platform to disseminate religious truth, though he was disinclined to support missionary proselytization publicly.[3]

There was a cordial relationship between missionaries of the London Missionary Society and a few local Hindu rulers who extended their monetary support to missionary educational establishments in Travancore. Enthralled by missionary education, these rulers even were willing to support English education unconditionally, though they were sceptical about missionary proselytization in the first place. For example, in December 1833, the Rajah of Travancore Rama Varma II visited a missionary school in Nagercoil of south Travancore. He specified that young women in Travancore were able to read, write and answer questions in both vernacular and English. He expressed that he was highly pleased with the missionary instruction given to his people. He even ordered INR 2000 to be paid to missionary of the London Missionary Society Charles Mault to make use of the money to construct a church.[4]

Despite cordial relationship between missionaries and a few Hindu rulers in Travancore, the continued missionary interference in the cultural life of the people generated discontent among the orthodox Hindus. While missionary encounters resulted in the reinvention of the self among the culturally conscious orthodox Hindus, marginalized women who had a soft corner for missionary propagated Christianity were willing to

be moulded by missionary teachings. However, it would be simplistic to assume that the religious teaching of missionaries was a separate watertight compartment that was completely delinked from the cultural project of colonialism. What is frequently referred to as the Western secular modernity was expanded further by certain Protestant missionary agencies, printing press, English education and colonial reforms (C. M. Bauman 2020, pp. 1–25). Most notably, colonial education not only attempted to cut the people off from their own culture and tradition but also made their own past esoteric to them as a history (Guha 1988, pp. 25–26).

The ways in which the defenders of tradition and advocates of modernity in the nineteenth century Travancore articulated their ideas need to be re-examined historically and critically. The recent interpretation that the nineteenth century struggles for attire decency in Travancore was a Shanar revolt and a seismic upheaval that 'demolished' all structures of power and dominance needs to be revisited in light of the theories related to tradition and modernity (Sheeju 2015).

## 2. 'Sheep in the Midst of Ravaging Wolfs'

Postcolonial studies have demonstrated how the globalization of Christianity, temperament of capitalism and ideologies of colonialism (re)shaped the discourse of tradition and modernity (Rudolph and Rudolph 1967, pp. 1–16; 29–49; Viswanathan 2000; D. M. Menon 2002, pp. 1662–67; Philip 2004, pp. 78–79 and 122–66; Barua 2015, pp. 12–13). These studies have suggested that there is an intricate interconnectedness between the industrial revolution, capitalism, imperialism, colonization and globalization of Christianity, which facilitated in the construction of modernity as a colonial form of knowledge (Philip 2004, pp. 1–28). The cultural transformation of the capitalist world economy, its widespread acceptance and the great intellectual, as well as cultural, synthesis of the long nineteenth century shaped the colonial and missionary institutional bedrocks (Wallerstein 2001, pp. 1–22). The colonial and missionary establishments endeavoured to modernize people either through mass conversions, also known as community movement or through character building education (Philip 2004, pp. 78–79 and 122–66). The missionary encounters with Hinduism had often underwired the connection between Christianity and British colonialism (Barua 2015, pp. 12–13). While a considerable number of the people from marginalized sections responded to missionary teachings positively, most of the orthodox Hindus challenged missionary discourse of modernity. The desirability of the marginalized sections towards Christianity in Travancore was partly prompted by the need to move away from the cycle of rigid social hierarchy, oppression and inequality and partly by the fact that Christianity allowed for their entry into a wider public sphere as individuals (D. M. Menon 2002, pp. 1662–67).

While missionaries as inquisitors were desperate to understand the cultural life of the people in Travancore, the colonial officials were reluctant from interfering into the cultural aspect of the people largely due to political reasons (D. M. Menon 2002, p. 1662). Some colonial officials and missionaries perceived that the aspects of tradition in Travancore were thematized and in some cases produced as instrument of rule by the dominant landowning class. The dominant sections, including Nairs, often perceived their cultural tenets as legitimate tools for political mobilization against colonial authorities and missionary politics. The orthodox Hindus and marginalized sections of Travancore equally tried to arbitrate new claims to status through well-established idioms of legitimation. When the Travancore state, through its decisive power over labouring populations, defended customs through official proclamations, its orthodox Hindus invoked the spirit of tradition to distinguish castes in public space and thus sustain their superior status claims, embodied in their dress code. Missionaries, on the contrary, used their religion to create a new identity among their converts (Jeffrey 1976, pp. 1–18; K. P. P. Menon 1924, pp. 95–99).

Missionaries perceived that Britain as a Christian nation had the highest form of civilization that had the capability to Christianize and 'civilize' its colonized people who were depicted as a 'race of extreme poverty'.[5] This demonstrates how the British missionaries

did not only define the social condition of the people through a materialist interpretation but also overlooked the cultural ethos through the ideals of Christianity. The discourse of modernity in colonial India was carried out through three distinct methods—proselytization by missionaries, legislative and administrative action by the colonial state—and dissemination of what the British called enlightened Western knowledge. These methods persuaded Indians to believe in the unworthiness of their way of life and to adopt the new forms of civilized and rational social British order (Chatterjee 1993, pp. 49–50 and 117–19; Viswanathan 2000; Dirks 2006, pp. 300–2).

This does not necessarily mean that the colonial government always supported missionary activities. A considerable number of colonial officials often expressed their apprehension that the missionary proselytization would be a serious threat to their political existence in India (Zupanov 2003, pp. 109–10). Nonetheless, as Britons and Christians, majority of the colonial officials had a sympathetic approach towards missionary proselytization.[6] Missionaries continued to develop a confrontational attitude towards Indian customs in order to intensify the process of modernization (Maclean 2008, pp. 125–26). The discourse of missionary modernity was a historical process wherein women missionaries attempted to modify the attire pattern of the marginalized women, advocating public decency and ensuring Christian civility. Missionaries were able to persuade the women who were more frequently the target of missionary reform initiatives. Missionaries asserted that they were all sharers in the great and blessed work of seeking to win souls for Christ from among India's women and indicated that women in Indian villages were exceedingly interested to interact with them.[7]

The itinerant Indian women preachers such as Yerraguntla Nagamma, Bangarappu Satyamma and others had a series of interactions with the fellow Indian women of the Telugu region in Madras presidency and introduced Christianity to them with a view to make superficial changes that are related to body, mind and soul (Taneti 2013, pp. 54–56). On the contrary, in some cases, there were instances of Indianization of Christianity to make conversion intelligible within the cultural meanings of the people. In South India, as missionary records indicate, women exhibited more penchants towards Christianity than men. In the case of converted Nadar women, missionary encounters through education sought to create a discourse of respectability that was contained, restrained, self-regulating and sexually chaste (Kent 2004, p. 240). When the colonial government was almost heedless about female education in the first half of the nineteenth century, missionaries used education as the first weapon to challenge the traditional gender stereotypes. Missionaries persuaded the converted women from the marginalized sections to work as local evangelists, teachers in missionary schools and henchwomen (Forbes 1996, pp. 32–60). The missionary education for the young women in Travancore propelled them to be critical of the traditional boundaries of gender, domesticity and orthodox femininity.

Missionary encounters with marginalized masses demonstrate how vernacular, and English education in colonial India were critical towards intra-gender inequality and caste rigidity (Oddie 1979, pp. 2–26) to create a consciousness for social upward mobility (Forrester 1980, pp. 3–23), and spiritual enhancement (Gladstone 1989, pp. 4–19). This is not to suggest that the missionaries themselves were completely fee from practicing social hierarchy at some level in both private and public spaces. In some cases, their writings and speeches emphasized more on equality before God than among people who were from dissimilar social backgrounds. Generally, missionaries advocated the principles of equality to condemn the rigid social structure of the Indian society and to reach out to potential converts (Viswanath 2014, pp. 15–19 and 40–99). Missionary criticisms were directed more against the Hindu beliefs, practices, rituals and customs than against labour extraction and occupational hierarchies (Kumar 2019).

The colonial officials and missionaries claimed that the Indian society was a culturally 'stagnant' society. As soon as the British established their political power in the second half of the eighteenth century, they realized the need for reforms and transformations in the Indian society. When missionaries attempted to disseminate Christian norms of equality

and decency among Hindus, the Travancore state and orthodox Hindus made initiatives to strengthen traditional attire pattern through commanding proclamations (Bayly 1984).

Thus, the marginalized women's encounters with missionary modernity and the missionary confrontation with the princely state of Travancore in the nineteenth century need to be situated within a broader context of colonialism, cultural hegemony and power dynamics. Colonial government and missionaries not only portrayed themselves as champions of respectability and instruments of change but also strived to win a reputation for common decency of life (Mateer 1871, p. 41; Devika 2005). For example, the struggle for attire decency was a missionary-induced struggle that provided the Nadar women a public space to advance their assertions of equality, justice, dignity and decency (Hardgrave 1968). In fact, from the 1820s until the first war of Indian independence of 1857, the direct missionary encounters with marginalized women were evidently noticeable where modernization endeavours had become the centre stage (Basu 1995).

It was a common practice in North India and in some pockets of South India that communities were structured around feudal landowning arrangements where the social status of a family was related to its ability to seclude and protect its womenfolk from potentially dangerous outsiders (C. A. Bauman 2008, p. 172). Travancore was, perhaps, one of the most culturally sensitive orthodox Hindu states, where the people practised their customs of orthodox Hinduism in a tangible manner. Travancore underwent a series of cultural conflicts after the political consolidation of the British and the introduction of Christianity by Protestant missionaries in the nineteenth century.[8] The Syrian Christians who trace their history back to AD 52 continued to occupy a highly honourable position in the state as merchants, officials, and politicians (A. V. Thomas 1974, pp. 90–91). The state and caste Hindus hardly had any major ideological conflicts with the Syrian Christian community for ages. Nonetheless, the Protestant missionary encounters that were critical of the Hindu customs, traditions and practices could possibly have prompted the Travancore rulers to reinvent their judicial, administrative and legislative establishments through political and cultural mobilizations.[9]

There was a strong nexus between religion and politics in Travancore in the nineteenth century. For example, the Travancore state was dedicated to Sree Padmanabha Swamy Temple, and the Maharaja of Travancore was considered the agent of the deity (Theertha 2004, p. 131). This indicates how Travancore state officially patronized the orthodox Hindu rituals and practices to defend Hindu faith (Bayly [1999] 2001, p. 162). While a large number of temples were under the direct supervision and control of the Maharaja, who was the supreme feudal lord, the feudal aristocracy controlled his authority and power (Jeffrey 1976, pp. 1–3; K. P. P. Menon 1924, pp. 95–96; Yesudas 1977, pp. 14–16). The Travancore state validated the very essence of caste as a social and cultural institution, thereby strengthening the social stratification (Mateer 1883, pp. 332–34). Subsequently, the Travancore rulers created a distinct provision for the administration to deal with matters of civil and criminal nature. To this end, orthodox Hindus such as Brahmins and Nairs played a crucial role in the principal and subordinate courts (Yesudas 1977, pp. 28–30).

While challenging this complex social and cultural stratification, women missionaries in Travancore argued that only conversion to Christianity would guarantee equality and social esteem to marginalized women. Missionaries believed that the marginalized Indian women would be able to experience decency, equality and respectability only when some of the controversial orthodox Hindu customs are discarded. As agents of change, missionaries endeavoured to disseminate what they called egalitarian values, offer a modern perspective of womanhood, question the deeply embedded orthodox Hindu customs and confront the attire pattern among the converted Christian women (Fleming 1992).

Thus, the colonial missionary encounter with the people of Travancore became tangible since the beginnings of the nineteenth century when London missionaries started working in Travancore encouraged by British Residents who were officially assigned there (Howard 1993, p. 19). It is significant to note that the practices of assimilation and adaptation in Travancore before the Protestant missionary confrontations were a common phenomenon

where local Hindu rulers allowed Christians, Muslims and Jews to freely practice their respective religions. Surplus lands were donated by the state for construction of churches, mosques and synagogues. Trade charters were issued to various religious groups to have control over ports and marketplaces. The intercultural interactions between different communities demonstrate that the Travancore state neither discriminated any religious groups nor intruded into the cultural realms of the people (Ouwerkerk 1994, pp. 47–48).

However, the arrival of Protestant missionary agencies created a new beginning in the cultural history of Travancore. A series of attempts were made to destabilize the social and cultural fabrics of the Travancore society through a superior Protestant religious imaginary. Missionaries had a cordial relationship with most of the British Resident commissioners. For example, Colonel Colin Macaulay and Major John Munro extended their generous support to missionary activities. Colonel Macaulay patronized the missionary proselytizing activities initiated by Ringeltaube, a German missionary of the London Missionary Society. In fact, Major Munro who had the distinction of being the most loved and venerated of the British Residents in Travancore assisted missionaries to intensify evangelization and modernization. The British Residents patronised missionaries, showed them much personal kindness and rendered substantial services towards educational works. Missionaries indicated that these Residents were idealistic, resourceful and extremely critical towards the privileged feudal aristocracy (Yesudas 1977, pp. 8–10). They expressed their gratitude for the help rendered by these Residents for the extension of Christianity in Travancore. Missionaries asserted that it was impossible for anyone to be more anxious for the promotion of Christianity in Travancore than Major Munro himself (Church Missionary Society 1819, p. 322; 1832b, p. 526).

The marginalized communities in general and the palmyra-climbing Nadars in particular attracted a great deal of missionary inquisitiveness, primarily as a subject of critical enquiry. Missionaries argued that a large number of Nadars were keen on pushing themselves forward from social and economic captivities (Aloysius 1998, pp. 17–18). They believed that the marginalized sections of the people would embrace Christianity when they are modernized through evangelization and Western education. However, the landowning communities such as Nairs challenged the controversial process of modernization, as it was seen as opposed to the established norms of Travancore Hindu tradition. The discourse of missionary modernity prompted the orthodox Hindus to exert pressure on the Maharajas of Travancore to detest missionary modernization. Subsequently, the Travancore state and orthodox Hindus expressed their discontent against missionary ideologies (Yesudas 1980), though some women from the royal family, including Gowri Parvathi Bai, supported missionaries and their religious discussions. Even though women from the royal families that belonged to the Nair caste received missionary education, they were disinclined to convert to Christianity for cultural and political reasons (Church of England Zenana Missionary Society 1881).

As overseers of land, the Nairs in Travancore enjoyed certain power and privileges under the pre-British policy. They were also known widely for their strong military training that helped colonial government to a great extent. Major Munro created two battalions of Nair sepoys and one company of cavalry headed by European officers to work as bodyguards and escorts to the royalty (Yesudas 1977, pp. 22–24). The Nairs, who were mostly feudal chieftains, had separate estates with distinct rights. Business transactions between Nairs and marginalized sections were conducted through an agent system in order to avoid pollution by physical touch. As feudal lords and warriors, the Nairs were hardly imprisoned for their crimes (Panikkar 1918). Even though Nairs demonstrated themselves as soldiers and officials who defended the rights of orthodox Hindus, they were classified as Sudras. They were classified as inferior in ritual status to Namboodiri Brahmins, non-Malayali Brahmins, Kshatriyas and Ambalavasis (the temple servants). As Sudras, Nairs appeared to have carried a degree of 'pollution' to Brahmin priests, Kshatriya Maharajas and temple premises. In Travancore, the notion of pollution was capable of transmission not only through physical touch but from a distance as well. Hence, Nairs

could approach but had no right to touch Namboodiri Brahmins; Pulaya slaves had to remain ninety-six steps away from a Brahmin; Syrian Christians could touch Nairs but were prohibited from inter-dining; and Ezhavas and Pulayas were considered as contaminating social groups (Jeffrey 1976, pp. 2–9).

The complex Travancore social settings impelled missionaries to challenge the social stigma of the marginalized women. The upper cloth struggle was interpreted not only as a sociocultural movement for a new identity with self-esteem but also was understood as an intense struggle for power between landowning Nairs and landless Nadars.[10] Missionaries believed that Nairs and other orthodox Hindus persecuted Christian converts from the marginalized sections for violating the deep-rooted Hindu customs and traditions. They specified that Nadars made repeated complaints against atrocities of Nairs in a respectful manner but such complaints were hardly entertained.[11] Missionaries maintained that on several occasions, Nadars in Travancore waited on the Diwan (the highest court official next to the Maharaja) of Travancore to put forth their complaints before him, but they were unceremoniously dismissed, because they wore the prohibited upper cloth. They insisted that, despite Nadars expressing their vulnerability, not a single government official, ranging from the Diwan to a peon, felt inclined to help them and speak a word on their behalf (See Note 11 above).

Missionaries indicated that Christian women were willing to stand up and fight back against what they called the arbitrary authoritativeness of the Travancore state and orthodox Hindus.[12] The *Madras Times* reported that Nadar women yearned for a decent and modest lifestyle precisely because going around half-naked was a moral and psychological hit to their feelings. They could not endure it but had to do so since they lived in a kingdom that promulgated the practice. The missionaries instructed the caste Hindus to either abandon their ancient customs or be willing to be civilized by the authority. When the orthodox Hindus refused to concede, missionaries vented their discontent that the customs were a foul blot upon humanity and should be obliterated so that civilization and advancement get a fair field. Missionaries proclaimed that the Maharaja of Travancore would never have cause to regret that he has in his kingdom a hundred thousand people so far advanced as to be ashamed of their condition (See Note 12 above). Consequently, missionaries portrayed Christians as sheep in the midst of ravaging wolfs. They believed that as long as the colonial officials had power in Travancore, they would have the liberty to carry out their proselytization in order to set people free from the state of degradation and ignorance (*The Scottish Congregational Magazine* 1860).

Missionaries contended that the Travancore customs laid down dress restrictions partly with a view to distinguish various social groups for occupational reasons and partly due to their distinctive cultural and ritual practices (Jeffrey 1976, pp. 57–59). It is significant to note that the attire patterns of men and women in Travancore were marked by simplicity. Covering the body was considered to be a mark of disrespect to both deities and to the customary practices. In some cases, women who dared to cover their body in front of the ruling elites were punished and their breasts mutilated by royal order (Pillai 2016). These instances indicate that it was almost possible to identify the caste of a person by dress and appearance (A. S. Menon 1978, p. 298). The women's attire pattern in Travancore did not change frequently and remained unaltered for centuries (Mateer 1871, p. 58). The custom of baring body was considered to be an integral part of the Travancore tradition, and a symbol of respect, as well as reverence. The Brahmin men showed their body as a mark of respect before the deities. The Nair women could cover their body with a particular style of cloth, which they were expected to remove before temple idols and people of superior castes.[13] The Syrian Christian women, who claimed a superior status to Nairs, wore a long-sleeved blouse. They were not expected to remove their cloth before high-caste Hindus.[14] Ezhavas were not only identified as the single-largest polluting caste but also were completely prohibited from carrying an umbrella or from wearing a shoulder cloth. Their women were forbidden from covering their body.[15] The social and cultural inequalities that were practiced in Travancore were also applicable to Nadars, even though community conversion

to Christianity had enabled them to escape some of the indispositions. Nadar women were prohibited from covering their body.[16] Similarly, the so-called slave castes such as Pulayas and Pariahs were forced to carry out the most arduous agricultural labour merely in return for food. Hence, missionaries claimed that conversion to Christianity was the only way forward for the holistic development of the marginalized communities. The vast majority of Christian converts in Travancore were from Nadar and Pariah communities that had become victims of patriarchal surveillance—a tangible violence against women (Jeffrey 1976, pp. 20–38).

The framework of deep-rooted patriarchal society in Travancore did not leave any stone unturned to keep women out of educational institutions. The women not only encountered issues that were closely linked to gender, identity, power and masculinity but also endeavoured to create an alternative discourse against their own men counterpart. The stereotypical portrayals such as 'men should never listen to women', 'a fifty-year-old woman should bow to a boy of five', 'a woman's sense comes too late' and so on proscribed women from education.[17] Similarly, a section of Nadar men in the Madras Presidency made sarcastic remarks against women's education suggesting that the patriarchal society was able to live well enough without educating its women for centuries. These men argued that imparting education to women would be detrimental to the patriarchal society, as they believed that education would make only 'troublesome' wives.[18] While challenging this strongly rooted patriarchal principle, converted women such as Krupabai Satthianadhan in Madras presidency strongly defended women's education. As has been specified in the census report in the 1860s, missionary education improved the social condition of women profoundly in the second half of the nineteenth century.[19] Furthermore, patriarchal society constructed the notion of dignity for women that gave men the ownership right over women's bodies within the framework of family. Similarly, husbands believed that their wives were their marital property. In many cases, women were classed with cows, mares, female camels, slave girls, buffalo, cows, she-goats and ewes. For example, the manner in which the Nadar men took up the struggle for attire decency suggests that women's bodies were perceived as men's property in one way or the other. It is equally important to note that the patriarchal society was preoccupied in controlling its women's bodies more than educating them. It was in this context, missionaries played an imperative role in creating a new cultural ecosystem where education and evangelization became tools for modernizing the marginalized women (Sarasvati 1888, pp. 50–60).

### 3. Discourse of Missionary Modernity: Proclamations, Negotiations and Contestations

While missionaries in Travancore strived to challenge the controversial attire pattern of the converted Nadar women, British Residents such as Major Munro sought to bring about a sweeping change in the clothing pattern in accordance with the spirit of Christian decency (Church Missionary Society 1819, p. 322). Munro's popular 1812 proclamation allowed converted Nadar women to cover their upper portion of the body for the first time in a manner that was in sync with the Christian women elsewhere in the world (Aiya 1999, pp. 236 and 526). The discourse of missionary modernity was thus initiated in Travancore through this first-ever official proclamation.

This proclamation specified that the orthodox Hindus had 'unnecessarily' interfered with the fundamental rights of the women who had embraced Christianity. It contended that the dominant caste men had prevented women from wearing their preferred cloth. It noted that those who become Christians should have the right to wear the shoulder cloth in accordance with the standards of decency befitting Christianity and in accordance with the customs prevailing in Christian countries. It emphasized that no opposition should be shown to Christian women.[20]

In reaction to Munroe's proclamation, the Travancore government issued a counter proclamation in 1814 allowing the converted Nadar women to cover their body with a short jacket as was worn by the women of privileged minorities such as Syrian Christians and Mappila Muslims. However, the Travancore state and the orthodox Hindus

did not allow the converted women to imitate the Nair women for cultural reasons (Hardgrave 1969, p. 60). The process of cultural imitation and the quest for upward mobility often created identity crisis between the defenders of missionary-advocated modernity and the upholders of Travancore tradition. The culturally sensitive Travancore state and the land owning Nairs appropriated this proclamation as a powerful tool for cultural assertion and political mobilization. The traditional attire of Nair women is the *mundum neriyatum*, a loose garment that roughly resembles the sari. This attire indicated the distinct cultural identity of the community (S. Thomas 2018, p. 38). When the converted Nadar women attempted to imitate the attire model of Nair women, Travancore witnessed a series of cultural conflicts. Missionaries and the British Residents encouraged the converted Nadar women to disobey the proclamation issued by the Travancore state. The Nadar women's continued replication of Nair women's attire led to the promulgation of yet another proclamation by Maharaja Venkata Rao in February 1829, which vindicated the Travancore custom and asserted that the disturbance was largely due to Nadar women's violation of the earlier proclamation. The state cautioned the Nadar women not to emulate the attire of Nair women (*Accounts and Papers of the House of Commons: East India, Annexation of Territory; Afghanistan, King of Delhi, Oudh and Travancore* 1859, p. 3). The rulers of Travancore were extremely critical of the Christian converts who were guided by missionaries. The Travancore administration, which had an incredulous attitude towards missionaries, ordered them to register their complaints directly with the Travancore government and not with the British officials (Strachan 1838, pp. 145–46).

Enraged by the missionary discourse of modernity, and the continued imitation of Nadar women, the Travancore state contended that women who belonged to Nadar, Ezhava and other marginalized communities were deliberately disregarding the Travancore customs by violating the established rule of law. The Travancore officials alerted that if such violations continue, it would be impossible to distinguish Nair women from other communities. The danger of touching the orthodox women based on the concept of purity-pollution triggered a series of discussion as to how the state should handle the issue responsibly. The state asserted that Nadar and Ezhava women should never imitate the attire pattern of Nair women, though it permitted Christian women to make and wear blouses based on the custom of Mappila Muslim women.[21]

The missionaries, however, continued to rouse the Christian Nadar women to fight for their right by breaking the outdated customary cultural practices including their attire pattern (Hardgrave 1979, pp. 162–63). Modernizing the marginalized women through education and evangelization became one of their powerful ideological tools (Strachan 1838, pp. 145–46; Church Missionary Society 1859, pp. 196–97). The continued cultural conflicts for attire decency resulted in violent clashes in various parts of Travancore such as Kalkulam, Yerenial and Valevengod (Strachan 1838, pp. 145–46). In many cases, the Travancore tradition and the missionary discourse of modernity appeared to be in direct conflict with each other. The reactionary attitude of the Travancore state and the subsequent inter-religious rivalry indicate how the anti-Christian outlook challenged the growth of missionary-propagated Christianity. Subsequently, missionaries appealed to the then British Resident Morrison, who, in turn, agreed to extend unconditional support to the Christian converts (Peter 2009, pp. 48–49; Ryland 1909, p. 5).

## 4. Politics, Power, and Identity in Post-1857

The struggle for identity, power and public decency took a new shape in the second half of the nineteenth century. The political scenario in the aftermath of the 1857 revolt augmented the power of the British colonialists. The official approval of the governor of the Madras Presidency was now required for the Travancore Maharaja to select his Diwan. It provided the British government a considerable amount of power in determining the political panorama of Travancore (Subramanian 2009, pp. 86–87). Meanwhile, the British Queen issued a proclamation in November 1858 emphasizing the need for religious neutrality, which pronounced that the government would refrain from interfering with

religious beliefs or worship of any of its subjects.[22] While the missionaries interpreted the proclamation as a threat to their proselytization activities, Nairs applauded the proclamation as they anticipated that it would give adequate power to local rulers to preserve their tradition, customs, conventions and rituals. Commenting on the changing political dynamism, Diwan Madhava Rao remarked that the mutual resentments between Nadars and Nairs were dormant for some time, but the Queen's proclamation renovated these feelings. While Nadars believed that the proclamation permitted them to infringe the existing Travancore rules, Nairs considered that it strengthened the right to protect Hindu traditions and customs (University of Calcutta 1872, pp. 241–45; Aiya 1999, pp. 235–36). Most notably, after the proclamation, the Madras Governor George Harris visited Travancore and restored the Maharaja with all the powers that the East India Company had unjustly taken away (Church Missionary Society 1859). Subsequently, the Travancore state and orthodox Hindus hoped that the Queen's proclamation would discontinue all its ties with missionary agencies, and thereby, missionary activities would cease to exist (Agur [1903] 1990, pp. 931–32).

After the Queen's proclamation, the Diwan Madhava Rao warned the converted Nadar women not to disrespect the rule laid out by the Travancore state. Retorting to the Diwan's admonition, missionaries proclaimed that the Travancore tradition was unsuited and unworthy of Christian decency (*The Evangelical Magazine and Missionary Chronicle* 1859, p. 729). Nevertheless, the principles of religious neutrality did not allow the British Resident William Cullen to defend missionary prerogatives. When missionaries made an appeal to Cullen to use his power to challenge the out-dated Travancore tradition, he expressed his discontent that the Christian converts were deliberately violating the established customary practices and forewarned that the converts would face consequences for their own imprudence (Church Missionary Society 1859). These remarks generated anti-Cullen sentiments among the missionaries who had persuaded the converts to defy the unsuitable customary attire practice. The missionaries instructed them to wear tailored blouses to cover their upper body like the Nair women (Mateer 1871, p. 278; Hardgrave 1969, p. 60). The tensions between the Travancore state and missionaries led to a series of violent struggles in Neyyantinkara, Kalkulam, Valevengod and Agastheswaram of Travancore in December 1858 (Mullens 1864, p. 105).

When the missionary modernity tried to challenge the 'incongruous' Travancore tradition, the culturally conscious orthodox Hindus endeavoured to reinvent their self through cultural (re)assertions. The state extended its unconditional support through proclamations with a view to defend the claims of those orthodox Hindus. Diwan Madhava Rao, in his 1858 order, notified that it was wrong on the part of Nadar women to violate ancient customs and antiquity without permission from the authorities. He threatened that those who violate the established customary practices would be severely punished.[23] The two-fold effect of this order was that Nadars had to respect the rule of the state and Nairs should not take law in their own hands. The then British Resident Cullen neither revoked nor expressed any disapproval of this order.[24]

Commenting on the colonial government's religious neutrality and the 1857 Travancore state order, missionaries remarked that the British Queen ensured social immunity to Nairs by giving more power to them to reassert their culture, dominance and power. Missionaries were unequivocal in condemning Resident Cullen for his failure to stop the violence against Nadars. They charged that Cullen did not bother to take any effective measures to treat the injured and punish the offenders (Church Missionary Society 1859). When the struggle for decency reached its peak in the 1850s, the then Secretary of State for India Lord Stanley directed Cullen not to recognize any exclusive distinctions repugnant to public morals. Lord Stanley also asserted that the proclamations made by the Travancore state were not only inappropriate for the modern age but also were in violation of the principles of dignity and decency (*Accounts and Papers of the House of Commons: East India, Annexation of Territory; Afghanistan, King of Delhi, Oudh and Travancore* 1859, pp. 3–4).

Consequently, the Travancore state proposed to abolish all restrictions on the converted Nadar women with some reasonable restrictions that did not allow them to imitate the attire of Nair women. The then governor of Madras C.E. Trevelyan directed both the Travancore Maharaja and the Resident Cullen to put an end to the controversial customs and traditions that violate human dignity (*Accounts and Papers of the House of Commons: East India, Annexation of Territory; Afghanistan, King of Delhi, Oudh and Travancore* 1859, pp. 4–5). The Christian Nadar women, armed with sympathy and support of the missionaries, demanded total abolition of the restrictions on covering their body (*Accounts and Papers of the House of Commons: East India, Annexation of Territory; Afghanistan, King of Delhi, Oudh and Travancore* 1859, p. 8).

Assisted by the missionaries, Nadars submitted a petition to the governor in the council of Madras Presidency titled 'The State of Travancore—The humble petition of the Nadar inhabitants of South Travancore' on 28 January 1859. The petition appealed to the governor to immediately intervene and help Nadars on behalf of the natural rights of women. It demanded same rights and privileges for their wives and daughters to decently cover their persons, as was enjoyed by women of orthodox Hindus in Travancore. It clarified that their claim had nothing to do with the pride of the attire, nor was it an invasion on the rights of others but was simply a requirement of natural modesty and one of those things that distinguishes human beings from brutes. The petition charged that it had always been the practice of the orthodox Hindus to suppress the petitioners. It informed the Governor of Madras that the stand taken by the Travancore state with regard to the exercise of right had been different at different times, in accordance with the recommendations of different British Residents in Travancore. The petitioners contended that the orthodox Hindus and the Travancore state had grossly and maliciously misinterpreted that the Queen's proclamation had driven away the East India Company and that all the rights and privileges that have been gained for Christian converts and for the so-called lower castes by the charitable help of the white men, were now nearing an end. The petitioners contended that as soon as this (mis)interpretation of the proclamation became generally known, orthodox Hindus assembled in crowds started beating the converted women and tore down their clothes. The petitioners impugned Resident Cullen for his insensitivity and irresponsible attitude towards Nadars. They contended that the officers of the Travancore state were not only caste minded but also portrayed Nadars as their chief enemies. The petitioners charged that instead of allowing Nadars to choose their path of decency, the Travancore state was using military power for the purpose of ensuring their degradation as imitation was interpreted as a state crime.[25]

Missionaries wrote extensively on the social plight of Nadars and their quest for upward mobility. In a letter dated 25 February 1859 to *The Madras Times*, John Cox, a missionary of London Missionary Society, argued persuasively that no slaves in Travancore had yet attained their freedom, and none of the Nadar and Ezhava women converts have ever worn an upper cloth. He claimed that he had always condemned the Travancore attire tradition because, when the cloth was worn without a jacket, it was, from its looseness, by no means a decent covering. He asserted that he had always encouraged all the Christian women to wear the jacket. He maintained that he had a number of meetings with Christians in 1858, and a more decided advance was made in the matter pertaining to women's decent attire. As a result of this, the women started wearing the jacket. Consequently, they were assaulted in public marketplaces and their jackets torn. He underlined that this issue was brought to the notice of the police, but the offenders were let off with not even so slight a punishment as a fine. He expressed his discontent that, since the reading of the Queen's proclamation, the ill feelings of Nairs have been far more markedly shown. He claimed that the predominant objective of Nairs was to create a revolution against missionary-advocated modernity. John Cox also stated that Nairs threatened the missionaries to stop preaching the gospel and not to open churches. In some places, he maintained, Nairs locked the churches, took away the keys and burnt the houses of Nadars in the presence of Travancore government officials (See Note 15 above).

Despite challenges and limitations, missionaries persistently declared that the moralizing effect of Christianity awakened the miserable Indian people to a sense of decency in their lives. Missionaries believed that all those who came under the spiritual direction of the missionaries were led to assume a different kind of attire that reflected what they called feminine modesty.[26] They argued that Nairs began to look down upon Nadars with jealousy due to their socioeconomic change through education. Missionaries expressed their apprehension that the Diwan of Travancore, from whose liberal education and enlightened views they had hoped for better things, failed to remove morbid feelings from the marginalized. They argued that the Travancore proclamations were intended to revive the obsolete and disgusting attire practice that compelled Christian women to expose their nudity to the gaze of the public (See Note 26 above). They sought to characterize the conflicts between tradition and modernity as valiant battles between the unsuitable conventional attire model that kept the marginalized in seclusion and the enlightened Protestantism that strived to promote the values of decency and respectability. The manner in which the land-owning Nairs lost control over the labouring Nadars and Ezhavas became a crucial point of debate. The struggle between tradition and modernity that had strived for continuity and change appeared to be gratifying to missionaries as they were able to use their religious ideals to modernize the marginalized women.[27] Consequently, missionaries were able to harvest souls, create wholesome homemakers and make reliable missionary assistants.[28] Nevertheless, the women who infringed the entrenched cultural practices faced a series of difficulties.

Missionary records indicate that those women who violated the Travancore tradition were sent to Trivandrum in order to undergo a severe punishment at the hands of the enlightened Travancore Maharaja. This, according to missionaries, demonstrates how the very act of covering one's own body was seen as a serious crime, almost as bad as killing a cow or touching a sacred Brahmin. Missionaries expressed their discontent that, despite unreasonable and unjustifiable actions of the Travancore state, the Maharajas and Diwans were protected and honoured by the most gracious Queen. They remarked that, while Nadars were kept in jail and exposed to all kinds of insults and tortures, Nairs were treated as respectable gentlemen and let out on bail and could thus hold their heads as high as ever.[29] Most importantly, the crux of the issue was that the Travancore state had decisive power over the labouring populations. This facilitated the state to arbitrate new claims to status. It was *that* power of the Travancore state that was at the centre of this series of historically specific conflicts. It was a question of holding authority and power. In some senses, the Nadars' struggle was not restricted to the idea of decency. On the contrary, it was a struggle for the same status and same privilege as the manner in which Nairs enjoyed.

While challenging the privileged status of the Nairs, missionaries persistently resisted what they termed the 'obsolete' Travancore tradition. They refused to allow their converts to submit to an observance so repugnant to all decency. This resulted in a series of riots between the sticklers for exclusive privileges and the unfortunate victims of an odious tyranny. According to missionaries, the consequences of these violent conflicts were disastrous. The British Resident's bungalow in Travancore and some of the protestant churches were burnt down, and the houses of the missionaries were destroyed. All those who were connected with the Protestant mission were obliged to proceed to Travancore to seek protection. Missionaries held a meeting with Resident Cullen in February 1859 to address the issues faced by the Christian converts. Yet, the meeting turned out to be an inconclusive one, as the Resident was disinclined to support the missionary point of view. Subsequently, a party of the Nair Brigade, under the command of Captain Daly, was sent to quell the Nadar–Nair violent conflict, which the missionaries termed the crying shame. Despite challenges, missionaries assured Christian women of their natural rights and persuaded them to infringe customs abhorrent to humanity through Christian notions of decency, equality and status (See Note 26 above).

## 5. Travancore Affairs, Communitarian Consciousness and Religious Identity

When the missionaries made appeals to Resident Cullen, he retorted that Nadars had broken the law of the state and therefore must be punished. Cullen sent Nair soldiers to put an end to the violent conflict and to reprimand the Nadars.[30] While commenting on the 'rebelliousness' of Cullen, *The Madras Daily Times* in its editorial 'Travancore Affairs' dated 1 April 1859 made a scathing attack on him stating that the religious persecution in Travancore was burning as fiercely as ever. The editorial remarked that a better man should be found to act as Resident in Travancore than Cullen. It specified that the best and most practicable way of accelerating the march of civilization was to create an enlightened native public opinion, honest enough to control the base tendencies of caste rigidity. It appealed to the people of Travancore to speak out plainly and vindicate their right to be considered worthy citizens of the British Empire in India (See Note 30 above). The manner in which missionaries and some British Residents endeavoured to modernize the marginalized women reveals how the colonial agencies in Travancore were keen on creating a Christian femininity.

While criticising the missionary modernity, some colonial officials remarked that the Christian converts were determinedly diluting the deep-rooted cultural practices of the orthodox Hindus by claiming equal status. Sir Mark Cubbon, one of the colonial authorities in Madras Presidency, contended in 1859 that the Christian women had forcefully imitated the Nair's attire pattern that was legitimately prohibited to them, and therefore, they must face legal consequences. Missionaries interpreted this controversial remark as an indication of the colonial government's irresponsible and worst attitude towards the marginalized communities and their newly adopted religion—Christianity.[31]

When the Maharajas and Diwans of Travancore expressed their discontent over the missionary intervention, a considerable number of people belonging to marginalized sections, consistently supported by the missionaries, were willing to be moulded by missionary ideologies. A substantial number of marginalized women started questioning and disowning the inferior cultural status imposed on them. The quest for equality and public decency propelled the marginalized women to transform their self in the light of the missionary modernity (Subramanian 2009, pp. 87–89). Missionaries used the idea of modernity as a potent tool to make an ecosystem for a profound cultural transfiguration. They strongly believed that their religion-driven modernity would create worthy and civilized British subjects (Devika 2005). It is significant to note that the British colonists and missionaries embarked on a cultural transformation mission where their predominant motive was to demonstrate themselves as cultural germ carriers who triggered the quest for freedom from social and cultural supremacy of orthodox Hindus (Smith 1958, pp. 450–51).

The letters exchanged between the Secretary of State for India, the Madras Governor, the missionaries and the British Resident commissioners in Travancore between March and June of 1859 reveal how the discourse of missionary modernity was explicitly expressed showing a sympathetic attitude towards the converted Nadar women (*Accounts and Papers of the House of Commons: East India, Annexation of Territory; Afghanistan, King of Delhi, Oudh and Travancore* 1859, p. 13). In a memorandum to the Maharaja of Travancore in February 1859, the missionaries argued that despite the abolition of slavery in 1855, Nairs failed to treat Nadar slaves in an appropriate manner. The British Resident Cullen, on the other hand, informed the Madras government that he had received a large number of complaints from Nairs that they had completely lost the services of their Nadar slaves on Sundays due to their conversion to Christianity. The Resident also questioned the missionaries for giving complete exemption to Nadars on Sundays from performing social and religious services to people from the so-called high castes (*Accounts and Papers of the House of Commons: East India, Annexation of Territory; Afghanistan, King of Delhi, Oudh and Travancore* 1859, pp. 7–15).

Those Nadar women who followed the practice of keeping their body uncovered before their conversion to Christianity had changed their attitude after their conversion by openly challenging the controversial attire pattern with unconditional missionary support. Consequently, Nairs contended that Nadar women did not only violate the state proclama-

tions but also used their Christian identity to question the deep-rooted cultural practice (*Accounts and Papers of the House of Commons: East India, Annexation of Territory; Afghanistan, King of Delhi, Oudh and Travancore* 1859, pp. 5–6). The Resident Cullen defended Nairs stating that the violence could have been averted if the Nadar-dominated neighbouring people from the neighbouring regions had not supported their caste men in Travancore and made counter-attacks on Nair villages (*Accounts and Papers of the House of Commons: East India, Annexation of Territory; Afghanistan, King of Delhi, Oudh and Travancore* 1859, pp. 6–7). Diwan Madhava Rao remarked that when Nadars took it upon themselves to infringe the state proclamations, Nairs considered that it was their moral responsibility to punish those who infringed the customs. He justified that Nadar women were attacked only when they appeared in public imitating the attire pattern of the Nairs in violation of the state rule and social customs (*Accounts and Papers of the House of Commons: East India, Annexation of Territory; Afghanistan, King of Delhi, Oudh and Travancore* 1859, pp. 8–10).

The continued struggles for cultural assertions and quest for decency generated intense debates among the lawmakers in London in the 1850s. The members of the House of Commons sought clarifications from the Secretary of State for India Lord Stanley. Even though Lord Stanley declined to give any pledge of interference, he termed the struggle for cultural assertion and the quest for decency a hypothetical case that needed a critical scrutiny.[32] Despite persistent appeals made by the Madras governors George Harris and Charles Trevelyan and the missionaries, the Travancore state asserted that the cherished Travancore tradition would never be terminated completely, as it would be against the rule of law. Yet, the state extended the idea of attire equality to Hindu Nadar women for the first time, and they were allowed to emulate the Christian Nadar women by preference. The 1859 proclamation of the Travancore Diwan reiterated that there was no expostulation to the Hindu Nadar women the wearing a blouse in the same manner as the Christian Nadar women.[33] In fact, this proclamation created a new model where both Hindu Nadars and Christian Nadars were deemed equal. It eventually proposed to abolish all cultural commandments perceived to be barriers for Nadar women and to grant them perfect liberty to meet the requirements of decency, with the reasonable restriction that they would not imitate the attire of Nair women (See Note 32 above).

As defenders of decency, missionaries denounced the traditional Travancore practice as a barbaric law and once again petitioned the Madras Governor in July 1959 to repudiate the Diwan's proclamations. They questioned the continued presence of Resident Cullen, for being hostile to missionary demands (Hardgrave 1969, p. 69). Despite reactions from the Travancore state and the British Resident Cullen, missionaries continued to use the struggle for attire decency to disseminate Christian notions of respectability and equality among the converted women. The persistent missionary pressure finally made Cullen to resign from all his official responsibilities as the Resident in 1860. Later, the Travancore state initiated a scholarship in memory of Cullen in the Trivandrum High School (Brown 1874, p. 502).

The struggles for continuity and change demonstrate how aspects of tradition and modernity in India were actually thematized and produced as a symbol of rule, power, privilege, status, decency and equality. The orthodox Hindus had incentives to argue on the basis of their tradition that became a well-established idiom of legitimation. The principles of tradition were used as potent political tools to challenge missionary modernity. The orthodox Hindus used tradition-centric proclamations to distinguish castes in the public space and thus maintained a somewhat superior status embodied in dress codes.

On the contrary, the proponents of change, most specifically the converted Nadars, Pariahs, Ezhavas, and Pulayas, were inclined to adopt Christian notions of respectability, public decency and equality. It would be simplistic to argue that the missionary modernity was the only ideology that guided the marginalized women to challenge the social and cultural hierarchy throughout the period under study. In certain scenario, leaders of marginalized communities such as Jotirao Phule, Muthu Kutty Swamy, Rettaimalai Srinivasan and Iyothee Thass Pandita in Madras Presidency, and in contexts unconnected

to the missionaries directly, also challenged the layers of hierarchy and power with an anti-orthodox communitarian consciousness. As an effective instrument, missionary modernity successfully challenged the power dynamics of tradition. Even though the Travancore state used tradition as a potent tool to confront missionary modernity, it could not dissuade the missionaries from creating communitarian consciousness among the marginalized communities in general and Christian identity among the converted Nadar women in particular. One of the crucial outcomes of missionary modernity was that it strengthened caste consciousness among Nairs and Nadars who came out with their own community organizations such as Nair Service Society and Nadar Mahajana Sangam, respectively, in the first few decades of the twentieth century. Similarly, the missionary modernity was instrumental in creating a new religious identity among the marginalized communities who converted to Christianity (Rudolph and Rudolph 1967, pp. 29–49).

## 6. Concluding Observations

The colonial discourse of missionary modernity pierced through the cultural sphere of the people through negotiation, contestation and resistance. It solidified communitarian consciousness and sharpened religious identity among the privileged, as well as marginalized, communities. As no-changers, the Nairs were able to uphold the age-old Travancore attire pattern. On the other, as pro-changers, the converted Nadar women were keen on adopting missionary perceptions of respectability, decency and equality.

While the quest for decency among the converted women was the intended consequence of missionary modernity, marginalized sections that were unconnected to missionaries also endeavoured to change aspects of their body to demonstrate their transfigured social and cultural status. The quest for modernity among the marginalized Nadar women did not only expand the process of religious conversions among other caste groups but also impelled them to be critical of their ascribed social and cultural hierarchy. However, the missionary modernity that attempted to offer a re-evaluation of the marginalized women's identity was limited largely to those who were either converted to Christianity or to those who adopted the principles of secular modernity disseminated by missionaries.

The unrelenting missionary sympathy for converted women indicates that the missionaries perceived those pro-changers essentially as a formative influence on next-generation converts. The well-established idioms of missionary modernity such as decency, equality and respectability were instrumental in creating a Christian femininity. On the contrary, however, orthodox Hindus assertively defended their attire pattern as an integral component of their glorious tradition. They also struggled to retain their traditional attire as a potential instrument for cultural assertion and political mobilization.

The persistent missionary encounters with the cultural life of the people impelled the Travancore state and orthodox Hindus to defend orthodox femininity through cultural assertions. The missionary modernity particularly among the marginalized women had a direct clash with the orthodox femininity through conflicts and confrontations. As potent players, women missionaries were efficacious in constructing a Christian femininity among the converted women in general and Nadar women in particular, for the first time, in the nineteenth century. The critics of missionary modernity, on the contrary, persuasively defended their venerable customary attire practice of Travancore in order to uphold their orthodox Hindu femininity. The discourse missionary modernity did not only impel the converted women to challenge their ascribed sociocultural status but also bolstered communitarian consciousness and religious identity among Hindus and Christians on an almost equal footing.

## 7. Limitations of This Study

A lack of vernacular voices from marginalized women—those Nadar women who embraced Christianity and women from other castes—is one of the serious limitations of this study. Further research is needed that needs to include oral narratives of both lettered and unlettered women who may have had mixed responses to missionary discourse of modernity. While some women could perhaps have sided with the missionaries and their idea of modernity by supporting the issues of the marginalized women, others could possibly have showed a critical attitude towards missionary modernity due to social, cultural and political reasons.

**Funding:** This research received no external funding.

**Conflicts of Interest:** The author declares no conflict of interest.

## Notes

[1] Council for World Mission Archives, Outgoing letters: India, 1822–1939, India General, Box no. 1–59, H-2133, Zug, 1978, 1913–1935, Microfiche, 5307–5349, no. 5324, Nehru Memorial Museum and Library (hereafter NMML), New Delhi.

[2] *The Friend of India*, No. 3, Vol. 1, 15 January 1835, Real 2863, Serampore, Calcutta, 1 January 1835 to 28 December 1837, NMML, *Friend of India* for the year 1835, Volume I, Serampore, MDCCCXXXV, *Friend of India*, Microfilm Real 2863, Serampore, Calcutta, 1 January 1835 to 28 December 1837, NMML, New Delhi.

[3] *The Friend of India*, No. 12, Vol. I, 19 March 1835, Serampore, Calcutta, 1 January 1835 to 28 December 1837, NMML, *Friend of India* for the year 1835, Volume I, Serampore, MDCCCXXXV; *The Friend of India*, No 42, Vol. I, 15 October 1835, Microfilm Real No. 2863, NMML, New Delhi.

[4] *The Friend of India*, No 18, Vol I, 20 April 1835, Serampore, Calcutta, 1 January 1835 to 28 December 1837, Friend of India for the year 1835, Volume I, Serampore, MDCCCXXXV, *Friend of India*, Microfilm Real 2863, Serampore, Calcutta, 1 January 1835 to 28 December 1837, NMML, New Delhi.

[5] Northcote to Lawrence, 15 August 1867, Lawrence Papers, Letters from Secretary of State, Vol. iv, No. 36; Also see (Gopal 1965, pp. 38–40).

[6] Indian Records of the United Society for the Propagation of the Gospel, Micro Film, Acc. No. 2137, S.P.G. 1, NMML, New Delhi.

[7] Council for World Mission Archives, Reports, 1866–1940, North India, Box No. 1-11 (H-2134) Zug. 1978, Box No. 2-1898, Microfiche, 6860–6906, no. 6866, NMML, New Delhi.

[8] *The Friend of India*, Vol. XXIII, No. 1158, 12 March 1857, 1 January-31 December 1857, Microfilm Roll No 2882, NMML, New Delhi.

[9] *The Friend of India*, 20 January 1859, 6 January 22 December 1859, Vol. XXV, No. 1255, Serampore, Calcutta, Microfilm Roll No. 2884, NMML, New Delhi.

[10] *The Madras Times*, 5 January 1859, *The Madras Times*, 5 January–28 February 1859; 1 March–30 June 1859, Microfilm Roll No. 903, NMML, New Delhi.

[11] *The Madras Times*, 9 February 1859, *The Madras Times*, 5 January–28 February 1859; 1 March–30 June 1859, Microfilm Roll No. 903, NMML, New Delhi.

[12] *The Madras Times*, 3 March 1859, *The Madras Times*, 5 January–28 February 1859; 1 March–30 June 1859, Microfilm Roll No. 903, NMML, New Delhi.

[13] *The Madras Times*, 10 March 1859, *The Madras Times*, 5 January–28 February 1859; 1 March–30 June 1859, Microfilm Roll No. 903, NMML, New Delhi.

[14] *Harijan*, Vol. I, No. 6, 18 March 1933, Harijan 11 February 1933–7 December 1935, Microfilm Roll No. 1312, NMML, New Delhi.

[15] *The Madras Times*, 7 March 1859, *The Madras Times*, 5 January–28 February 1859; 1 March–30 June 1859, Microfilm Roll No. 903, NMML, New Delhi

[16] *The Friend of India*, 10 February 1859, 6 January 22 December 1859, No. 1258, Vol. XXV, Serampore, Calcutta, Microfilm Roll No. 2884, NMML, New Delhi.

[17] Hilda Lazarus, 'Woman's Work for India's Women: Work in the Zenana as a Missionary Agency,' *The Harvest Field*, Addison and Co., Madras, May 1893, pp. 425–27.

[18] *The Madras Mail*, 12 March 1869.

[19] Report on the Census of the Madras Presidency (1871), Report on the Census of the Madras Presidency, 1871, with Appendix containing the results of the Census arranged in standard forms prescribed by the Government of India, by W.R. Cornish, Sanitary Commissioner for Madras, Vol. I, Madras, Printed by E. Keys, at the Government Gazette Press, 1874, pp. 111–2.

[20] Order of Her Highness Parvathi, whilst Colonel Munro was the Resident, given at Quilon in the year of the Quilon Era 988 = AD 1812 19 Margali to the Sarvadi Karyakartas of the districts of Trivandrum and Negauttangarey. See Note 11 above.

[21]　Order of Narayana Meneven, Diwan Peishkar, whilst Colonel Newall was Resident, dated Quilon year 999 equal to AD 1824. See Note 11 above.

[22]　East India (Proclamations), Copies of the Proclamation of the King, Emperor of India, to the Princes and the Peoples of India, of the 2nd day of November 1908, and the Proclamation of the Late Queen Victoria of the 1st day of November 1858, to the Princes, Chiefs and People of India, India Office, 13 November 1908, Arthur Godley, Under Secretary of State for India (London: Eyre Spottiswoode, 1908), pp. 2–3.

[23]　No. 20, Memorandum from Major H. Drury, Assistant Resident in charge, Travancore and Cochin to the Chief Secretary to Government, Fort Saint George, dated Trivandrum, 28 September 1859, No. 85, Proceedings of the Madras Government, Political Department, for the month of November 1859, Govt. of Madras, Pol. Dept., G.O. No. 666, dated 12 November 1859, National Archives of India (NAI) New Delhi.

[24]　Order of Madhava Row, the then Diwan, General Cullen, being Resident, nearly two months after the publication of the Proclamation of Her Majesty, Quilon Year 1034, AD 1858, December 27, proclamation to all people, *The Madras Times*, 9 February 1859, *The Madras Times*, 5 January–28 February 1859; 1 March–30 June 1859, Microfilm Roll No. 903, NMML, New Delhi.

[25]　'The State of Travancore'—The humble petition of the Shanar inhabitants of South Travancore to the Right Honourable the Governor in Council, 28 January 1859. See Note 11 above.

[26]　*The Madras Times*, 28 January 1859, *The Madras Times*, 5 January–28 February 1859; 1 March–30 June 1859, Microfilm Roll No. 903, NMML, New Delhi.

[27]　Report of the Committee, delivered in substance to the annual meeting held on 1 May 1832, (Church Missionary Society 1832a, pp. 60–61).

[28]　Letter from G.E. Phillips to Rev. A.K. Legg, London Mission, Quilon, South Travancore, India, 14 March 1935, Council for world Mission Archives, Outgoing letters: India, 1822–1939, India General, Box no. 1-59, H-2133, Zug, 1978, 1822–1939, Microfiche, 5307–5349, no. 5342, NMML, New Delhi.

[29]　*The Madras Times*, 14 March 1859, *The Madras Times*, 5 January–28 February 1859; 1 March–30 June 1859, Microfilm Roll No. 903, NMML, New Delhi.

[30]　*The Madras Daily Times*, 1 April 1859, *The Madras Daily Times*, 5 January–28 February 1859; 1 March–30 June 1859, Microfilm Roll No. 903, NMML, New Delhi.

[31]　*The Madras Daily Times*, 19 April 1859, *The Madras Daily Times*, 5 January-28 February 1859; 1 March-30 June 1859, Microfilm Roll No. 903, NMML, New Delhi.

[32]　No. 21, Order Thereon, 12th November 1859, No. 666, *Proceedings of the Madras Government*, Political Department, for the month of November 1859, Govt. of Madras, Pol. Dept., G.O. No. 666, dated 12 November 1859, NAI, New Delhi.

[33]　*Proceedings of the Madras Government*, Political Department, for the Month of November 1859, Govt. of Madras, Pol. Dept., G.O. No. 666, dated 12 November 1859, NAI, New Delhi.

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
