# Peer review of "Religions, Women and Discourse of Modernity in Colonial South India"

_religions, doi:10.3390/rel13121225_

Round 1

Reviewer 1 Report

The paper's main argument is that the binary tradition-modernity is inadequate to understand the complexities of the transformation of gendered identities in colonial south India, particularly as seen in the breast-cloth controversy of Travancore, which roiled the southern tip of India from 1800 to the 1860s.  It's not clear to me what positive argument the author is making an alternative.  I think the author is arguing that missionaries have been seen, and saw themselves, as agents fostering the transformation of identities from traditional (and "backward") to modern (and "civilized"), but the actual processes of change were very complicated and contested.  This is not a particularly original contribution to scholarship on colonialism.  The author's copious footnotes indicate familiarity with the relevant scholarship, and also access to and interest in primary sources from the period, but the paper doesn't  contribute much that is  new to existing scholarship on gender and colonial modernity, or to analyses of the breast cloth controversy.  The framing of the data/evidence is at once extremely broad (postcolonial critiques have demolished any illusions that colonialism or missionaries benefited Indians), and extremely narrow (basically recounting or describing what articles in the Madras Times or the text of a governmental order say). 

It is clear that a lot of research has gone into this article.  Moreover, the breast cloth controversy is such a rich and complex site for analysis it is certainly possible to generate from it new insights into the transformation of Indian society under colonialism.  Thus, I see considerable promise here for the paper's further development.  Here are some suggestions for doing that:

1) identify what exactly is problematic about the term "missionary modernity" and unpack that systematically for the reader.  Clarify what you want to argue with or about this material. 

2) review in a more systematic way the existing scholarship on the breast cloth controversy and indicate what aspects of it you are building on, where you disagree, and what interventions in interpretation/analysis or what new data you are bringing into the conversation.  Hardgrave, J. Devika, Kent, and Thomas have all discussed this at length.   I'm not sure if this paper is based on new data, or whether it revisits material from the period that has already been looked at.  This review doesn't have to be long, but without it, it's  hard to see what new moves this paper is making. 

3. The lengthy discussion of colonial modernity and the place of missionaries within it needs refining.  For example, on page 3 (lines 67-85),it is unclear whether the critics of colonial modernity here are from the past, or present day critics. Similarly, are the "defenders" of modernity, people in the past or the present?  I got  excited thinking that T. Murali was a painter from the 19th century who had actually documented the controversy (which would be quite a find, if such a painter existed!), but then googled him to find out he's a contemporary painter.  My confusion stemmed in part from this section's lack of clarity about the point of view from which various critiques are being launched.  As an aside, a whole paper could be written on contemporary mobilizations of the breast cloth controversy by artists, historians, Indian Christians and activists to assert community identity - and this author is may well be well positioned to do that.

4. Sweeping generalizations need to be eliminated, or refined.  For example, on page 4 (lines 124), the author writes that relations between local HIndu rulers and missionaries were cordial, but is this really so across all the regions of INdia, and across time? 

5. "Feminisation of Christianity" in the south is not well explained.  Are you arguing that there were more women involved in Christian missions in south India?  That women were more frequently the target of missionary reform initiatives in south India?  What evidence is there that more women and than men were interested in Christianity in the first place? 

6. p. 7 - Oddie, Forrester and Gladstone are all characterized as "recent" studies on missionary encounter, but these are now almost fifty years old.  How does Sonja Thomas's work complicate this? 

7. p. 8 - here is where a sub-theme emerges that I think may be the paper's thesis, or a secondary thesis that could be made more prominent - that the missionaries thought they were promoting "equality, justice, dignity, decency, and power" (with the exception of the latter, universal moral qualities, being mostly immaterial in nature) but actually what was at stake (from the author's point of view, from the historical actors' point of view?) were material-political gains (i.e. challenging the hierarchy of the caste system, which was visibly instantiated in dress and also perpetuated economic exploitation).   Gladstone I think makes this case, but it could be made again in conversation with people like Viswanathan's work on missionary "uplift" of "pariahs", and recent scholarship on colonial modernity and capitalism. 

8.  In the first few pages of the article, it would be helpful for the author to step back and describe what was going on with the breast cloth controversy so as to give readers unfamiliar with this material a purchase on the analysis of the series of proclamations that ensued. 

Author Response

My responses to the first reviewer’s comments

  1. Based on the comment, I have identified the challenges/limitations/problems of ‘missionary modernity’. I have tried to differentiate missionary-advocated modernity from the modernity of colonial government.
  2. I have reviewed the existing scholarship on the theme under study and indicated what aspects of it I am constructing on, where I differ, and what interferences I am bringing into the debate. 
  3. Based on the suggestion, I have tried to demonstrate to what extent my argument is different from Robert Hardgrave, J. Devika, Eliza F. Kent, and Sonja Thomas.
  4. I have tried to critically examine the material from the period that has already been used extensively. 
  5. I have now specified that the defenders/critics of modernity are used here mostly in the context of the cultural past of the Travancore.
  6. Based on the suggestion, I have now avoided sweeping generalisations. Rather, I am redefining the earlier sentence ‘that relations between local Hindu rulers and missionaries were cordial.’ I am trying to be more specific geographically. This is not to suggest that the ‘cordial relationship’ is applicable to all the regions of India, and across time.
  7. I have now defined the term ‘feminisation of Christianity’ in the context of south India. I use this phrase to argue that women missionaries were keen in inculcating the idea of English, vernacular, and technical education among women in south India. I have also used the term to demonstrate how marginalised women were more often the focus of missionary ‘reform’ initiatives in the place under study.
  8. Based on the comment, I am now using the term ‘postcolonial studies’ instead of ‘recent studies’ to refer to the works of Geoffrey Oddie. I have incorporated the works of Rupa Viswanathan and Sonja Thomas to substantiate my argument.
  9. Based on the suggestion, I have examined two inadequately studied narratives (‘static’ femininity and ‘decent’ femininity) that are related to what is commonly referred to as ‘the breast cloth controversy.’ This, I think, will give readers new perspective on the theme of the discussion.

Reviewer 2 Report

A well written and thoughtful article on the moot research questions of the creation of a ‘self’ amongst the Christian converts of lower castes, power and authority struggles between the state of Travancore and the dominant castes. Although the ‘breast cloth’ controversy of early modern southern India is well known amongst gender and cultural historians, the author brings refreshing insights to the old question by tracking it alongside the ‘modernity-tradition’ debates. It’s value is in how it moves away the central gravity of the issues to questions of ‘power’ and struggle for control and authority amongst the Nairs and the state of Travancore especially in the post rebellion period. So, in my opinion, yes, the article is worthy of publication in your esteemed journal.

My main concern is about the structure of the article which impedes the reader’s enjoyment of the main arguments and analysis. Currently it is divided into 7 sections of uneven lengths (this imbalance is alright by me but the number of sections can be pared down to 4-5 in my opinion). The missionary modernisation discourse and its complex trajectories are traced along with the postcolonial critiques of it in the beginning in two sections which are superbly executed. Candidate has used the latest theoretical interventions without neglecting to mention and engage with the older historiographies. Additionally, the author culturally, temporally and geographically locates the debates in colonial Travancore in the third section allowing the reader to know the context of the deeply mired tradition-modernity nexus of state sovereignty, legitimation of economic and cultural control by dominant Hindu castes and legitimacy with Protestant missions in this particular region of British India and how ideas of decency, respectability, social status and social mobility, economic imperatives are linked to attire for Travancore women.

 The first two sections can usefully be just one section as the author is challenging the idea of looking at the breast cloth controversy purely in the modernity-tradition paradigm. But I would also recommend a collapse of section three (Locating the Debate to Travancore) and 4 “Encountering the ‘Other’” as the themes are in a continuum here and too many sections often do a disservice to the arguments of the author by distracting the reader. There are repetitions of ideas too which can be pared.  In fact, section 4 is still very much a continuation of the previous section as it is ‘contextualising’ the 'other'. I am not at all doubting the rich insights and signposting that section 4 does in terms of letting the readers know about the long historical trajectory of body, shame, respectability discourses and how the dominant castes kept the Shanars/Ezhavas in their place and how they enjoyed the power and privileges this unequal economic relationships (instrument of rule) traversed and melded with social and cultural ones such as attire of the women of these castes. But it seems logical to collapse some of these sections into two rather than 4 sections and see where the arguments can be made tighter and also compact.

I found section 5 “Fashioning the Gendered Other…” highly relevant and the particular negotiations and contestations are brought out extremely well through the three major proclamations – 1812, 1814 and 1829. I can already see how I would use this in my teaching of UG courses! This can be retained as a stand alone section but again, I would recommend that the author retitles and makes the section heading clearer with the ‘proclamations’ inserted in it. After all the major struggles that are taking place are a direct consequence of state action being expressed through their muscular power – the Proclamations. The next section 6 titled “Politics, Power and Proclamations” is misleading the reader. Proclamations are simply continuing but the crucial change here is the periodisation. It is after 1857. So the author could usefully retitle some of the sections using the chronological language such as ‘pre rebellion politics and power’ and also ‘post rebellion power and politics’? This strategy might actually enhance the author’s own argument that the quest for identity and power takes new shapes in the post 1857 period which currently remains hidden or obscured.

Section 6 is paradoxically the most powerfully articulated one in this article. It’s a gem but still quite rough. I say this because the author talks about the quest for the creation of a ‘self’ on behalf of the marginalised groups of Travancore but we never see the agency of these rebellious groups spoken about on their terms. Rather their agency is subsumed as something determined constantly by missions and missionaries even after their conversions. I find this a troubling way of dealing with colonised people’s agency. I would recommend the author reads Jeffrey Cox’s amazing work on the politics of contemporary historical scholarship in Asia (in the journal Women’s History Review) wherein he berates both subaltern and postcolonial scholarship for constantly neglecting to see the agency of converts in their own terms. I see this pattern emerging in this article too wherein no matter what the rebel does – to create a self for herself she is still not enabled or empowered because her consciousness is constantly read as determined (corrupted?) by other forces/discourse be it the dominant caste or state or missionary. The author might try reading works by gender and historians of women who have addressed the agency of women converts in other cultural contexts of India such as Maharashtra (Sharleen Mondal, Padma Anagol in her book [chapter one is on Indian Christian women or her WHR article], Meera Kosambi and others). I realise that the agency of converts is not the core argument here but the whole article seems quite deterministic and reads their agency in linear terms. Maybe the author can try putting a disclaimer that s/he is not concerned with the agency of the converts? This may do the trick. 

This study is a valuable addition to the growing literature on religion, state and society and ought to be published after some revisions.

Author Response

Please find attached herewith my responses to the comments of the second reviewer

Reviewer 3 Report

Dear contributor,

This was a wonderful paper and I enjoyed reading it. The only reason (if I may be excused for it) I marked the need for more references in your work, is that I find a lack of voices in your text. While you have provided an excellent overview of the debates and discussions around modernity and decency, using the breast cloth controversy as an arena where the struggle for modernity, caste, and gender took place in 19th cent Travencore, I miss what women themselves felt about the issue. In this whole feminizing of Protestantism in Travancore that you discuss, there must have been plenty of women's voices: Nadar women converted to Christianity, and also non-convert women (who may have wanted to convert or not), who may have witnessed this epistemic struggle playing out on their bodies. I know that the basic data you have worked with is archival (and secondary), and you have indeed done a tremendous job by demonstrating the differences between the Resident and missionaries. But this is also the beauty of the missionary archive (in contrast to colonial records). The whole 'soul' saving thing, brought to the fore individual 'voices' of the 'suffering', that in the form of oral history has gained traction in the postcolonial period as a powerful research method. Even though 'voices' within missionary texts are deeply mediated and overwritten by missionaries themselves masquerading as those trying to be saved expressed in what were heartwrending accounts, missionaries had very deep contacts with the local community. And these conflated accounts nevertheless constituted 'voices' that would provide significant depth to an article like yours (though you have summarized this from newspaper reports of the time in a paragraph).

Take for example the whole gender-sexuality debate that the citing of these voices would bring up. While you have outlined the caste 'humiliation' factor implicit in Nadar women forced to uncover their breasts that hit them psychologically, what would be the gender-sexuality issues implicit within the act of covering? Covering in the context has equal ramifications as uncovering. The act of covering locates the dignity of female sexuality and motherhood in the breast, and locates this dignity as part of a husband or father family property and ownership rights over the bodies of women within the family. More than the father, covering definitely connotes the property rights of the husband -- the Nadar man in this context. It is the violation of this property right that is imbricated with dignity, decency and shame. This brings other debates about women's pleasure to the forefront. What about earlier written vernacular 'voices' or essays on this? It could form an important part of your analysis, especially since you have skirted the entire sexuality issue which is somewhat central to the topic itself.

This would be my 50 cents! I look forward to seeing this article published. Its a great contribution!

Best wishes!         

Author Response

Please find attached herewith my responses to the comments of the third reviewer

Round 2

Reviewer 1 Report

This is a tough call because the author has clearly made substantive changes to the article.  I still however, find myself struggling to understand what the central argument is and how the author is marshaling evidence to support that argument.   One recurring issue is the excessive use of scare quotes, which seem to have increased in the revision process. These make it very difficult for the reader to know how we are to understand the phenomenon or concept or issue enclosed in quotes - as socially constructed, as contested, or as terms that other people (scholars and historical actors) use but in ways this author finds inadequate or problematic ('modernity,' 'liberate,' 'greater security.').

Author Response

Dear Editor,

Thank you so much for your suggestions and comments to improve the quality of my paper.

I have incorporated all the changes suggested by academic editors

I have gone through the draft carefully and edited it accordingly.

As suggested by the academic editor, I have also written a short paragraph on the limitations of this paper.

I will be happy to hear from you if you have any queries for me to address.
